# Addressing Sustainable Development: Promoting Active Informed Citizenry through Trans-Contextual Science Education

**TBM Chowdhury \*** , **Jack Holbrook and Miia Rannikmäe**

Centre for Science Education, University of Tartu, Tartu 51014, Estonia; jack@ut.ee (J.H.);
miia.rannikmae@ut.ee (M.R.)
\* Correspondence: tapashib@ut.ee

**Abstract:** This article seeks to identify the role of science education in promoting an active, scientifically literate, citizenry ready to address sustainable development goals as envisaged by the United Nations (2015). In so doing, a conceptual model is put forward to address citizenry development, extending beyond an informed scientific and technological decision making ability and encompassing constructive activities addressing sustainable development at the local, national and global level. The operationalisation of the model builds on an initial student-relevant, societal issue-related contextualisation involving STEM (science, technology, engineering, mathematics) while focusing on developing science conceptual learning. The model extends to not only considering socio-scientific issues, but seeks to promote trans-contextualisation beyond the school setting, seeking to raise awareness of an active informed citizenry, related to environmental, economic and social sustainability. The components of active informed citizenry are described and a trans-contextual science teaching example based on the model is put forward in this article.

**Keywords:** active informed citizenry; education for sustainable development (ESD); global citizenship; science education; trans-contextualisation

---

## 1. Introduction

A growing concern is addressing sustainable issues within school teaching-learning activities, thus seeking to promote Education for Sustainable Development (ESD) [1]. This is considered a key to achieving target 4.7 emphasising Global Citizenship Education [2]. Through emphasising a link between promoting scientific literacy and addressing societal engagement, both individually and collectively, there have been calls for educational reforms [3], and the need to take into consideration paradigmatic challenges in science education, related to preparing students for societal changes [4]. In fact, it has been suggested that the existing teaching-learning approaches are '*unsustainable*' [5].

Within school science education, the inclusion of socio-scientific issues (SSI) has emerged as an important educational construct [6]. This enables emphasis on preparing students to participate as citizens within a democratic society, well-acquainted with scientific conceptualisations and their engagement in society issues involving science, or the wider perception of STEM (interrelating the scientific conceptualisations with technological ideas, engineering procedures and mathematical enhancements) [7]. While SSI, in the literature, is seen as controversial [8], ill-structured [9], focussing on socially embedded issues [10], requiring an understanding of the nature of science [11], its inclusion in science education is being recognised as playing a role in promoting global citizenship [12,13].

One important goal of SSI is to effectively address attributes promoting future citizens through science education [12,14,15]. In addressing this goal, the literature recognises the need to draw attention

to students acquiring personally responsible, participatory decision making skills. Besides promoting an appreciation of opportunities to address future careers [16], such skills draw attention to the needed roles to be played by the society, especially in the areas of environmental protection, health, and social adhesiveness. Nevertheless, it has been pointed out there is a danger of underestimating students' ability to identify constructive measures and assume a sense of responsibility to take personal initiatives towards collective engagement in order to resolve SSI and sustainable development issues [17]. Furthermore, it is suggested that insufficient research exists linking an individual's decision-making through SSI, within the school setting, to the potential of student activity and community involvement outside the classroom [18]. This points to a potential gap between the desired SSI informed decision making outcomes and students' preparation through gained socio-scientific attributes as a desired citizen. In fact, it has even been pointed out [19] that ineffectiveness in socially embedded science instruction within the classroom (e.g., about vaccination, or nuclear explosion) can result in activities that have a negative socio-economic repercussion (e.g., anti-vaccination campaigns, siting nuclear plant concerns).

Not teaching controversial issues in the science classroom (for example, with respect to sexual and reproductive health, HIV, teenage/unintended pregnancy) can impact on the ability of students to make well thought out future decisions regarding their personal life [20]. This raises a concern that including socio-scientific issues in teaching without a vision of the need to prepare students to make informed decisions, as and when required, can result in failure to achieve the expected learning [21], or even in some cases, such as uncontrolled, irresponsible alcohol or tobacco consumption, leading to counter-productive outcomes [22]. Uncertainty in seeing how to deal with any SSI aspect, stems from its multi-disciplinary, complex nature. This is illustrated when reflecting on uncertainty in situations that can be considered as have the potential to lead to chaos when common agreed intent is lacking (as per the cynefin framework) [23].

The aim of this article is to address a proposed need for school education to go beyond developing the individual and focus on socio-scientific decision making as a preparation towards handling complex situation by promoting a desired citizenry (Although the science education literature tends to use the terms 'citizens', 'citizenship' and 'citizenry' interchangeably with a similar meaning, this article intentionally uses the terminology 'citizenry' to mean a collected group of citizens who have a commonality in their social purpose, as opposed to individual citizens, and where citizenship is conceptualised as a status of these individual citizens.), able to strive, in particular towards attaining sustainable development within the society. In so doing, the article proposes the need to go beyond SSI decision making and introduces the need for a trans-contextual society impacting stage, still within an education through science approach [24]. The significance of this article is two-fold. First, it lies in the conceptualisation of a desired 'Active Informed Citizenry'. Subsequently, it puts forward the need to operationalise this through a motivational school science education learning model [23], in which an additional model component pays attention to ways to address sustainable development goals [2] beyond the classroom.

## 2. Importance of Promoting Citizenry for Sustainable Development of the Society

The education role, in addressing citizenship in the 21st century and its relationship with education, needs to go beyond the individual and engage at the society, or even the global, level [25] and be in line with the 2030 agenda for sustainable development [2]. The agenda draws attention to the need to promote the knowledge and skills necessary to achieve a sustainable lifestyle, recognise and protect human rights, promote gender equality, establish a culture of peace and non-violence, conceptualise global citizenship and develop an appreciation of cultural diversity and of culture's contribution to sustainable development. This is suggested as a step towards citizenry.

While school curricula mention the need to develop students' capabilities to function as citizens [26], it is suggested there is a further need to develop students' capabilities to become 'good' citizens, based

on their collective actions towards a better world [16]. This 'citizenry' role is not only limited to the local, or national level [27]; it is recognised as a necessity for preparing societies at a global level [12].

## 3. Role of Science Education in Promoting a Desired Citizenry

Typically, science taught in school has largely been a cognitive endeavour [28], but developments over the last 20–30 years have emphasised the need for a wider focus (e.g., the STS movement followed by the emphasis for the inclusion of SSI – [10]). Within this wider focus, the need for a changing perception through addressing the function of the society in the total teaching-learning process is being recognised [29].

In science, or STEM education, there is increasing attention to social inter-relationships [30], especially inter-disciplinary [31], and even towards a trans-disciplinary focus [32]. Within school science curricula, the focus on enhancing scientific literacy is well established [33,34], implying the involvement of student learning as going beyond cognitive capabilities and encompassing social and career aspects [16]. This is intended to lead to better informed citizens. Nevertheless, it is argued that emphasising only informed citizens is insufficient. A desired citizenry needs to be participatory, allowing citizens to be able to play an active role in the resolving of issues within the society [35]. Such a scientific literacy shift, pays more attention to promoting a socially responsible and competent citizenry, in line with sustainable development goals. It can be expected to go beyond solely active citizenship [36], and informed citizenship [37], and embrace the wider aspects of science education at a global, or international level. Recent studies [38] imply that such a more contemporary science education can be seen to be contributing to this.

## 4. Conceptualising Active Informed Citizenry within Science Education

The concept of an active informed citizenry is intended to give an 'all-embracing' idea of citizens acting together who are meaningfully informed, educated not only to play a role at a national level, but also actively prepared to embrace wider, global issues, recognising these may also impact at a national, or even local level. Science education can be expected to play a role in such an endeavour noting that each component in the expression–active informed citizenry–has its own interpretation. Thus:

- the term 'active' indicates a willingness and preparedness to participate and engage in science-influenced personal, societal and even political demands. Besides gaining knowledge and conceptualisation of scientific issues, students need to engage in meaningful science-influenced activities [39]. Curriculum emphasis can be given to enable the learners to observe experts, or even teachers, while engaging in an action, practice the skills in a specific context, take responsibilities to plan and organise the actions, engage in critical evaluation of the plans and actions from the teacher and the peers during the action and afterwards;

- the term 'informed' relates to achieving a level of scientific literacy relevant to the engaged school curriculum and out-of-school experiences. The characteristics of scientifically literate individuals are suggested as possessing a profound knowledge and understanding of science for problem solving, critical thinking, risk and benefits of science [40], identifying evidence, drawing conclusions, communicating and demonstrating conclusions based on science [41], advocating a central role for scientific knowledge and perceiving scientific literacy for a social benefit [34];

- the term 'citizenry' indicates the plurality of citizens of a particular region, recognising an educational need to prepare citizens with a sufficiently meaningful scientific behavioural activity and engage in socio-scientific issues as a collective citizenry. From a science education point of view, a collective citizenry can be visualised [35] as, for example,

    - organise responsible groups;
    - write and distribute letters and petitions to the respective authority;
    - boycotting products and practices from a socio-scientific point of view;
    - take initiatives to promote positive citizenry behavioural change;

- take the initiative for resolving ethically fair, science-influenced issues, and
- promote innovative solutions for local, or even, global problems.

The term 'active informed citizenry' actually appears in the literature e.g., [42,43], although it is identified more at a national level, focusing on consideration of the duties and rights of a citizen. The term, as used in this paper, goes further and additionally encompasses collective, globalised aspects. It strives to recognise that a more idealist view of citizenry suggests both the need for the development of informed citizens (here in a scientific literacy sense) and active citizens (in the sense of playing a meaningful role). These are subsequently brought to bear on the SSI-derived, consensus decision so as, potentially, to drive a collective society development leading to an active, informed citizenry.

In conceptualising a desired citizenry model, to be achieved through science education, Figure 1 highlights different attributes that can be grouped, contributing to the desired active, informed citizenry (AIC) target.

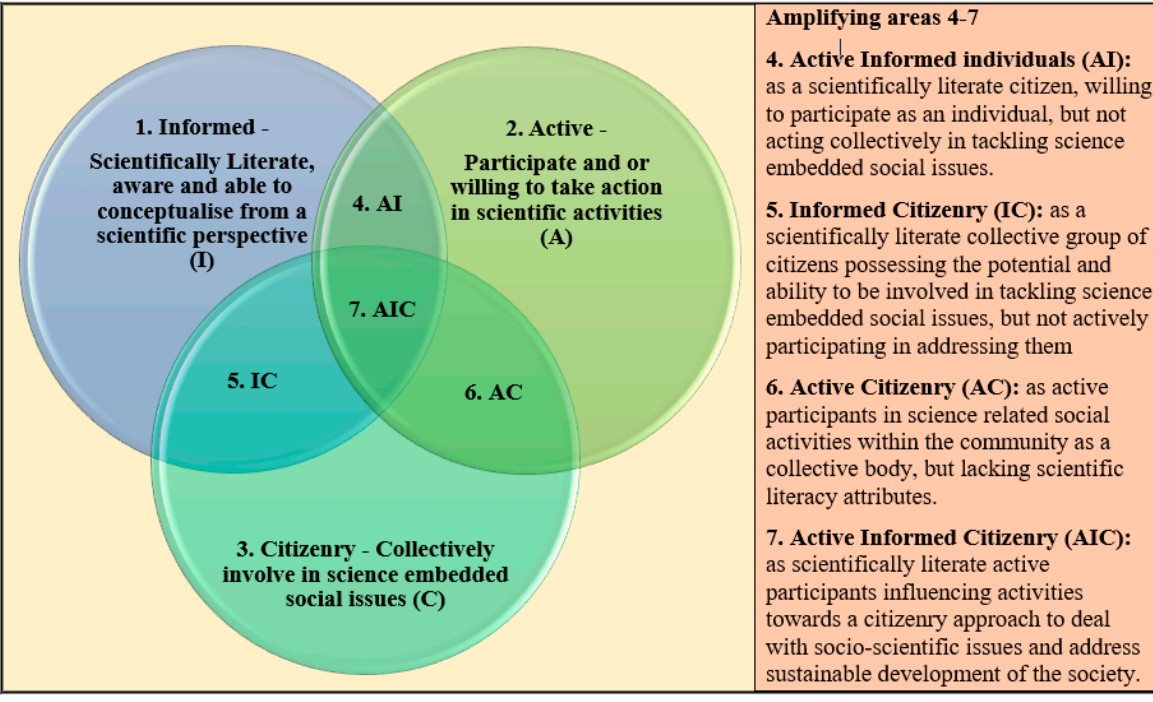

**Figure 1.** Illustrating Active Informed Citizenry and associated attributes to be derived through science education.

Figure 1 identifies attributes to be developed though science education, building on a SSI baseline (1, 2 and 3), going beyond active informed individuals (associated attribute 4), or informed citizenry (attribute 5), or even active citizenry (attribute 6) so as to better address sustainable development goals. In developing an active informed citizenry, there is a need to go beyond an interdisciplinary learning approach and build a wider learning platform linking school to the society.

## 5. Trans-Contextualisation Promoting Active Informed Citizenry through Science Education

In this paper, trans-contextualisation is envisaged as moving from a within-school learning setting to a wider development platform within the society where big changes in social behaviour can be involved. In so doing, it strives towards building on students' informed decision making in the school setting to extend this to a wider societal audience, thereby seeking to stimulate justified active informed actions and enable sustainable development within the society.

The rationale and the particular characteristics for a continuum within science education, leading to promoting trans-contextualisation, can be illustrated through a 4 stage process. The first 3 stages are based on an 'education through science', as opposed to a 'science through education' philosophy [24], leading to a contextualisation, de-contextualisation and then re-contextualisation model [23].

*Initial Contextualisation Stage*: Using a familiar and relevant social context, portrayed as a societal issue or concern, involving a science component, to initiate learning that can more motivationally relate to science. To address the inadequate relevance of science education to students' daily lives, potentially resulting in lack of interest and motivation towards science learning [43], the contextualising of the science learning within a social context, seeks to focus on:

(a) motivating and identifying relevance for students through a familiar social issue or concern;

(b) incorporating students' perspectives on the relevance of the context towards establishing educational value;

(c) determining the level of students' prior science and science-related learning related to meaningfully addressing the issue from a science conceptual perspective;

(d) recognising and identifying students' need to gain further, or more in-depth, science conceptual learning to be able to address the issue.

*De-contextualisation Stage:* De-contextualising, from the social setting, to address the need to acquire relevant science competences. The de-contextualising of the learning, involves students acquiring new science, driven by a 'need to know' based approach. In involving the students in science, or science-related [STEM] learning, the teacher may utilise a structured, guided, or open inquiry learning approach, as befitting the students' prior learning. In this phase, teaching is expected to mainly focus on:

(a) recognising that science learning is needed to address the social issue;

(b) appreciating how to address the required science learning;

(c) promoting scientific conceptualisation and skills, through meaningful challenges, and

(d) enabling students' self-actualisation through the learning process.

*Re-contextualisation Stage*: Applying the acquired science learning from the de-conceptualisation stage. The gained science conceptualisations, alongside meaningful consideration of other social factors, are involved within a group interaction to undertake socio-scientific decision making through argumentation. The goal is to resolve, in a class consensus manner, the socio-scientific issue identified in the initial contextualisation stage. A major outcome is expected to be enhancing students' ability to discuss, debate, make informed decisions on social issues, based on a scientific background [13].

The teaching focus within the re-contextualisation stage emphasises:

(a) applying acquired scientific conceptualisation to address a social issue;

(b) developing transformative competences in line with the goals of education (e.g., argumentative reasoning, justified decision making, role playing, etc.) within the social context, and

(c) promoting justified and scientifically informed decision making skill in a consensus, democratic way.

*Trans-contextualisation Stage*: Applying the science learning, within a sustainable development arena, to promote engagement in social issues, having a science component. This relates to everyday life both within, and even outside, classroom considerations. This stage seeks to enhance awareness and involve active participatory approaches to controversial issues of a local, national, or global nature. It further seeks, through collective actions, to stimulate a sense of commitment to undertake unified actions beyond the school, leading to active informed citizens.

The added stage is based on a concern that there is a perceived lack of attention, linking an individual's decision-making (re-SSI) to societal activity outside the classroom. This stage is intended to encourage students to take constructive post-consensus, action measures, thereby gaining a sense of responsibility for taking personal initiatives. Such initiatives are instigated by recognising the need to promote collective engagement beyond the classroom in order for active and informed operations within the society [17,18]. The trans-contextualisation stage seeks to engage students in transferring their learning from an educational institutional environment to the wider environmental, economic, social (at a local, national, and/or global) arena, thus addressing the sustainable development of the society. The trans-contextualisation stage builds on the theory of collective activism [34,43,44], which recognises actions that may include, for example,

- changing one's own behaviour (for example, recycle, reduce, reuse, increase energy efficiency);
- proposals for innovative solutions to social problems;
- encouraging active participation in volunteer initiatives;
- developing ways to seek to persuade and educate others (such as through exhibitions, social network activities, blogs), or
- stimulate the operations of lobby groups.

## 6. An Example to Illustrate Trans-Contextual Activities within a 4-Stage Teaching Approach

The following figure (Figure 2) illustrates a trans-contextual stage, extending beyond the 3 stage approach based on an 'education through science' philosophy [24,44,45]. The example is based on a suggested grade 10–11 science (chemistry) level topic - thermoplastics and thermoset plastics.

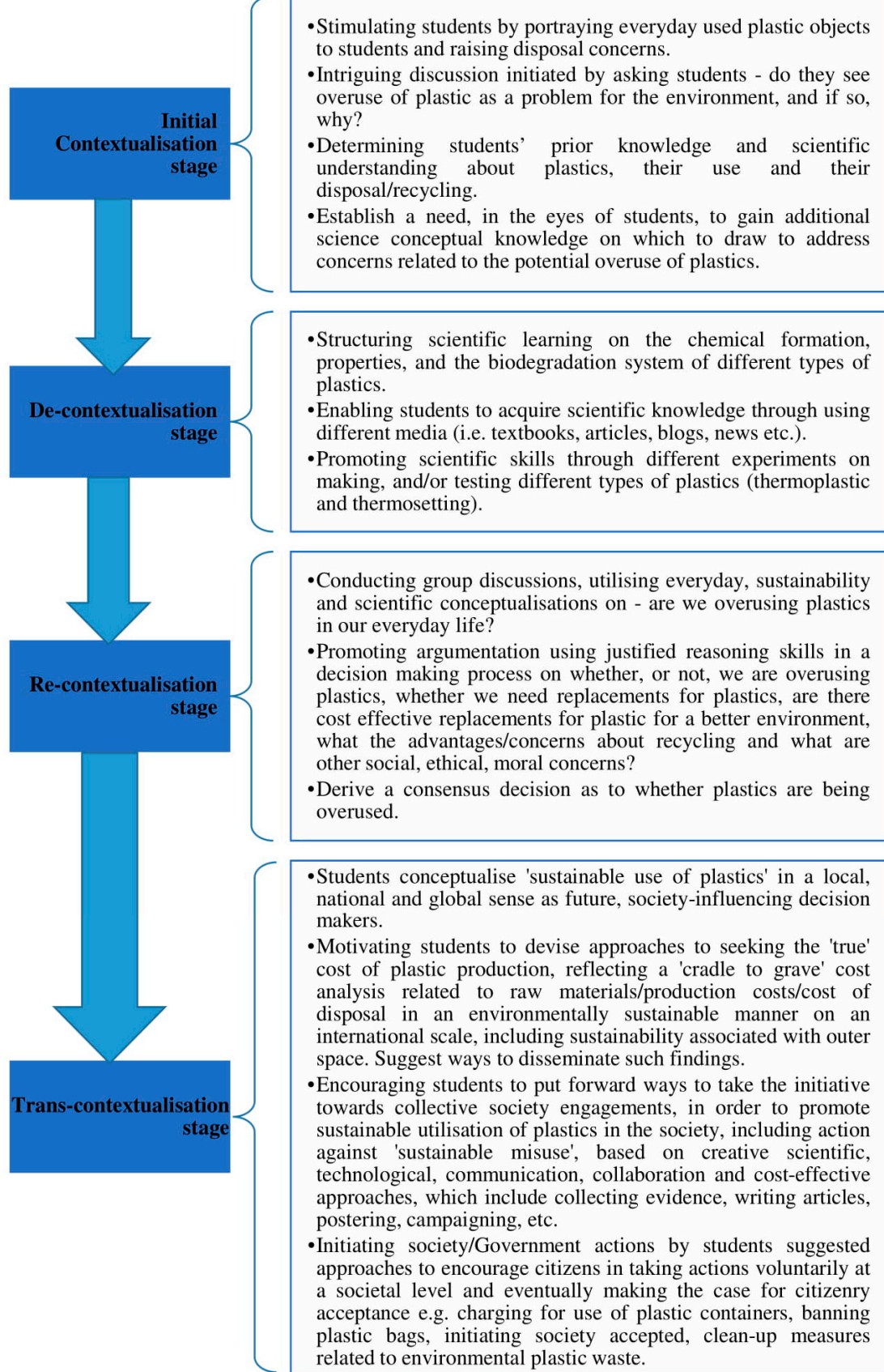

**Figure 2.** Illustrating exemplary activities within each stage of the 4-stage model.

## 7. Conclusions

The paper focuses on promoting a wider goal for science education, by adding a trans-contextualisation component, important with regard to the sustainable development of a society. It puts forward the role of science education as not only developing citizens as individuals, such as through promoting SSI, but going further to develop an active informed citizenry, thereby stimulating a willingness by the society to engage in sustainable development activities.

**Author Contributions:** The conceptualization and methological approach, T.B.M.C. and J.H.; Checking and validation, T.B.M.C., J.H. and M.R.; Writing—the original draft preparation, T.B.M.C.; Writing—review and editing, J.H., T.B.M.C. and M.R.; Visualizations, T.B.M.C. and J.H.; Supervision, J.H. and M.R. All authors have read and agreed to the published version of the manuscript.

**Funding:** This research received no external funding.

**Conflicts of Interest:** The authors declare no conflict of interest.

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
