# Peer review of "Addressing Sustainable Development: Promoting Active Informed Citizenry through Trans-Contextual Science Education"

_sustainability, doi:10.3390/su12083259_

Round 1

Reviewer 1 Report

The manuscript advocates the combination of traditional science education with socio-scientific issues. The authors extend an existing model (developed by two of the authors) with an additional stage: trans-contextual. The manuscript is conceptual, not empirical. Still, the discussion presented is important and interesting.

The example in table 1 shows how this can be done in practice.

The manuscript is generally well structured and well written.

Please note that capital letters should be used for proper nouns only, not for highlighting words or phrases.

  1. Introduction

L27 “Target 4.7”, propose to use “target 4.7 of the United Nations sustainability goals”

L56 “siting a nuclear plant concerns”: something is wrong here. Please rephrase. Maybe “siting nuclear plant concerns”.

  1. An example to illustrate trans-contextual activities within a 4-stage teaching approach

Table 1 is more a figure. It needs to be redrawn/rewritten. Some problems with formatting

  1. Conclusion

It is important to emphasize that discussions should be based on a scientific approach, and not on popular opinions.

Author Response

Dear Reviewer,

Thank you so much for your precious comments. We acknowledge your suggestions as very important and made changes accordingly.

Thankfully yours,

Tapashi

Reviewer 2 Report

The paper is well thought out and is an introduction to the literature that is currently available.  The reference list is adequate and again well researched.  I do not detect any problems with the paper.

Author Response

Dear Reviewer,

Thank you for your acknowledgement to the paper and your precious time to review it. We really appreciate your gesture.

Yours,

Tapashi

Reviewer 3 Report

Comments/Suggestions for Authors

The logical structure of the paper needs to be improved:

For example, The importance of promoting the Desired Citizenry and the role of education in preparing the desired citizenry cannot be addressed without first defining the characteristics of the Desired  Citizenry.  So there should be a separate section after the Introduction to conceptualize who this ideal or desired citizenry would be like before discussing his/her importance and the kind of education necessary for actualizing his/her potentials.

Missing Discussion:

Also, it is not enough to provide the conceptual framework for educating the Desired Citizenry as you do with the stages and the Venn Diagrams. You need to follow up the design with a paragraph discussing and defending/justifying the logical order and importance the stages from 1-7.

Typographical or grammatical text edits:

Line 31: The term "unsustainable" as used in the context creates an amphiboly or ambiguity problem. It could mean:

a) Not suitable for Sustaining the environment etc. or 

b) Not likely to last for a long time.

So the authors should clarify what they mean to avoid the problem

Line 64. "This paper addresses" ... is more appropriate than "This paper addressing..."

Line 65. It "focuses on..." is better than It "focus on..."

Line 67 It "introduces..." is better than It "introduce..."

Line 70 It "puts..." is better than It "put..."

Line 88 "Developments have" is better than "Developments has"

Author Response

Dear Reviewer,

Thank you for your generous and thorough review, we really appreciate it. Here I am trying to explain and provide the possible changes according to your reviews.

1. The logical structure of the paper needs to be improved:

For example, The importance of promoting the Desired Citizenry and the role of education in preparing the desired citizenry cannot be addressed without first defining the characteristics of the Desired  Citizenry.  So there should be a separate section after the Introduction to conceptualize who this ideal or desired citizenry would be like before discussing his/her importance and the kind of education necessary for actualizing his/her potentials.

Explanation: In section 4, line 104, we tried to give the explanation of a desired citizenry. But now I understand, it has to come earlier before discussing the role of the science education to promote the desired citizenry. We are reviewing the article to best fit the suggested edit.

2. Also, it is not enough to provide the conceptual framework for educating the Desired Citizenry as you do with the stages and the Venn Diagrams. You need to follow up the design with a paragraph discussing and defending/justifying the logical order and importance the stages from 1-7.

Explanation: Thank you for pointing out the gap, we are giving a better discussion now based on your suggestion.

3. Line 31: The term "unsustainable" as used in the context creates an amphiboly or ambiguity problem

Explanation: The term was used by the reference cited in the sentence, we put it in italic form with inside quotation mark. But now we understand we have to explain the term for the reader to understand better. 

4. Thank you so much for the grammatical suggestions.

We sincerely thank you for the wise advices. 

Yours,

Tapashi